# The role of restrictive abortion laws on modern contraceptive use in Sub Saharan Africa

**Maxwell Tii Kumbeni**[1,2]*, **Marit L. Bovbjerg**[1], **S. Marie Harvey**[1], **Chunhuei Chi**[1], **Jeff Luck**[1]

**1** School of Nutrition and Public Health, College of Health, Oregon State University, Corvallis, Oregon, United States of America, **2** Ghana Health Service, Nabdam District Health Directorate, Nangodi, Ghana

* tiimax2@gmail.com

## Abstract

Previous evidence found that restrictive abortion laws were associated with high incidence of unsafe abortions and abortion-related complications. But few studies have assessed the association of restrictive abortion laws with contraceptive use patterns. Our study examined the role of restrictive abortion laws on contraceptive use among women in Sub Saharan Africa (SSA). We performed secondary data analysis using the most recent Demographic and Health Survey data along with the 2022 abortion law classification data from the Center for Reproductive Rights (CRR) for 31 countries in SSA. Based on CRR classification, abortion restriction was categorized into three, ordinal levels; broadly liberal, moderately restrictive, and highly-restrictive laws. We conducted multivariable, multilevel logistic regression analysis to estimate odds ratios (OR) and 95% confidence intervals (CI). The analysis also included multivariable imputation by chained equations to account for missing data. A weighted sample of 453,195 women of reproductive age (15–49 years) were included. Thirty-nine percent and 49% of women lived in countries with moderately-and highly-restrictive abortion laws, respectively. Women in countries with moderately-and highly-restrictive abortion laws were 0.62 (95% CI: 0.58, 0.65) and 0.88 (95% CI: 0.83, 0.93) times as likely to use modern contraceptives, respectively, compared to women in countries with broadly liberal abortion laws. However, in countries that had policies that supported adolescents to access contraceptives, women overall were more likely to use contraceptives. Liberalizing abortion laws and implementing policies that support women's autonomy in contraceptive decision-making could enhance contraceptive use among women in SSA.

## Introduction

Abortion laws are legal statutes that prescribe the circumstance under which a pregnancy may be intentionally terminated [1]. Historically, abortion laws were introduced

**Data availability statement:** Data underlying the findings of this manuscript has been added to the submission as a Supporting Information file.

**Funding:** The authors received no specific funding for this work.

**Competing interests:** The authors have declared that no competing interests exist.

because abortion was considered a sin and these laws were meant to punish and act as deterrent to individuals receiving and providing abortion services. Abortion laws were also meant to protect women and fetal lives because abortion was dangerous and many women were dying from abortionists [2]. But since legal abortion procedures and medications are much safer now, the intent of current abortion laws can only be for punitive purposes or protect fetal life over that of women's lives. Although prosecution of unsafe abortions that cause complications still exist, abortion laws are mainly used against those receiving and/or providing safe abortion services [2].

Legality of abortion around the world often ranges from broadly liberal (abortion permitted upon request or on economic grounds) to highly restrictive laws (abortion permitted to only save a woman's life or prohibited altogether) [1]. Abortion legality in many countries is also tied to gestational limits and certain circumstances such as rape, incest, and fetal anomaly [1]. The abortion laws in Sub Saharan Africa (SSA) were mostly inherited from the colonial era. Whereas many of the European originators of these laws have shifted to more liberalized laws, the majority of SSA countries still use the old restrictive laws [2]. As of 2019, more than 90% of women aged 15–49 years in SSA lived in countries where abortion laws were highly restrictive, permitting abortion access only when the woman's life is at risk [3].

The effects of restrictive abortion laws on women's reproductive health are profound. Such laws are associated with higher rates of unsafe abortions [3,4]. While the legality of abortion significantly influences it safety, it does not affect the frequency of abortion occurrence [4]. For instance, the global abortion rate remains consistent at 40 per 1,000 women of reproductive age annually, regardless of whether abortion laws are liberal or restrictive [5]. However, the safety of these procedures varies greatly – fewer than one percent of abortions are unsafe in countries with liberal laws, compared to 31% in those with restrictive laws [4]. In SSA, the Guttmacher Institute estimates around 33 unsafe abortions per 1,000 women of reproductive age each year [3]. Unsafe abortion remains a major contributor to maternal morbidity and mortality in the region [4,6]. Additionally, research shows that restrictive abortion laws are associated with poor mental health of women [7,8], adverse infant and child health outcomes [9–11], and long term socioeconomic hardship for families [12–14].

Despite the documented negative impact of abortion restriction on reproductive health of women, few studies assessed the association of restrictive abortion laws with contraceptive use, and reported mixed findings [15–18]. Some studies suggest that abortion restriction may be associated with lower contraceptive use, especially in countries with highly restrictive abortion laws, where access barriers may limits individuals to obtain contraceptives [16,19]. In contrast, other research indicate increase in contraceptive use to prevent unintended pregnancy, in response to highly restrictive abortion laws [17,20]. The mixed findings suggest that broader structural, cultural and socioeconomic factors play a significant role in shaping contraceptive behavior beyond legal framework of abortion [21–24].

Research on abortion restriction and contraceptive use has been conducted mostly in high income countries [15–18]. These finding may not be applicable in SSA because of the major sociocultural and healthcare systems differences between SSA and high

income countries. Therefore, abortion restriction could impact women in SSA differently especially on the use of modern contraceptives. It is common for SSA countries with highly restrictive abortion laws also to have policies that discourage contraceptive use [25,26]. The average prevalence of modern contraceptive use in SSA is lower than the global average (29% vs. 44%) [27]. Therefore, assessing the role of abortion restriction on contraceptive use would be essential in understanding their complex relationship and would offer key information on sexual and reproductive health policy and programing.

In this study, we used the global abortion law classification data from the Center for Reproductive Rights (CRR) and individual level data from the Demographic and Health Surveys (DHS) to estimate the association of restrictive abortion law with contraceptive use in 31 SSA countries.

## Materials and methods

### Data sources and study design

The primary data source for this study was individual-level secondary data from the most recent Demographic and Health Surveys (DHS) – 2012–2021. DHS conduct cross sectional surveys every five years in many low-and-middle income countries using a standard questionnaire across countries. Details of DHS sample design, stratification, and sample weights is publicly available. [28] We also used the 2022 global abortion law (country-level) data from the Center for Reproductive Rights (CRR) [1], the 2022 countries adolescent contraceptive legislation data from the Countdown to 2015 for Maternal, Newborn & Child Survival [29], and the current health expenditure (CHE) as percent of gross domestic product (GDP) data from the Global Health Expenditure Database [30].

The Center for Reproductive Rights is the leading source of global data on reproductive right laws, and policies. The center tracks and updates changes in abortion laws and penal codes across all countries in the world [1]. Abortion laws are categorized based on provisions in national statues, legal regulation, and court decisions. CRR uses legal experts in constitutional, international, and human rights law from various countries to review each country's law as they apply to reproductive health and rights. The Countdown to 2015 Maternal, Newborn & Child Survival is a global network that track, stimulate and support country progress towards achieving Millennium Development Goals (MDG) 4 and 5, now part of the Sustainable Development Goals (SDG) 3. They track data from over 75 countries across the world including many countries in SSA. Data were collected on several country-level indicators including contraceptive legislations. The Countdown analyses are guided by a conceptual model developed by a working group of members from the Countdown, the World Health Organization(WHO), the World Bank, the GAVI Alliance and the Global Fund. Details of data collection and analyses can be found at the Countdown [29]. Details of the Global Health Expenditure data is also publicly available [30].

### Study setting and participants

A total of 31 countries in SSA were included in the study. Countries were excluded if they had no available data in DHS or their DHS data was restricted or the data was 2008 or older. The cut off point of 2008 was arbitrarily chosen. Furthermore, countries whose current abortion laws preceded their recent DHS data were excluded from the analysis because it was not appropriate to apply the law retrospectively to older data. Detailed flow chart of the inclusion and exclusion criteria is found in S1 Appendix.

The study population were women of reproductive age (15–49 years). All women who were pregnant or were declared infecund at the time of the surveys were excluded from the analysis because they were likely not using contraception for reasons other than fertility control.

### Variables

**Outcome variables.** The outcome variables were use of modern contraceptives and use of long-acting reversible contraceptives (LARC)/permanent methods. Modern contraceptives were male and female sterilizations, intra-uterine devices (IUD), subdermal implants, hormonal pills, injectables, male and female condoms, emergency contraceptive

pills, patches, rings, diaphragms, vaginal spermicides, standard days methods and lactational amenorrhea. Our definition of modern contraceptive use was adopted from the 2015 WHO Department of Reproductive Health and Research, and the United States Agency for International Development (USAID) technical consultation team [31]. LARC/permanent contraceptives included male and female sterilizations, intra-uterine devices (IUD), and subdermal implants. Both outcomes were analyzed as binary measures.

**Exposures.** The primary exposures were country-level abortion law and adolescent contraceptive legislation. Abortion law was classified into three ordinal categories (i.e., broadly liberal, moderately restrictive, and highly restrictive abortion law) based on prior classification by the Guttmacher Institute [3]. Countries that permitted legal abortion upon request without women needing to provide any reason, or permitted legal abortion on socioeconomic and health grounds were classified as having broadly liberal abortion laws. Countries with laws that permitted legal abortion on health grounds only (i.e., whether physical or mental health or both) were classified as having moderately restrictive laws whereas countries that permitted legal abortion only to save the life of a woman or banned abortion altogether were considered as having highly restrictive abortion laws.

Adolescent contraceptive legislation was categorized into three ordinal levels (i.e., no legislative support, partial legislative support, and full legislative support for adolescents to access contraception). No legislative support referred to countries that had no legislation in place that allowed adolescents to access contraception. Partial legislative support referred to countries where legislations were available to support either married adolescents to access contraception without spousal consent or unmarried adolescents to access contraception without parental consent. Full legislative support also referred to countries where there were legislations that allowed both married and unmarried adolescents to access contraception without parental or spousal consent. This classification was adopted from the Countdown to 2015 for Maternal, Newborn & Child Survival [29].

The other exposure variable was current health expenditure (CHE) as percent of gross domestic product (GDP). CHE as percent of GDP was measured using the standardized parity power purchase approach. CHE as percent of GDP corresponded with the year of survey for each country.

**Covariates.** The covariates included duration of abortion law, age, number of living children, marital status, place of residence, education level, wealth index, health insurance coverage, religion, visited by family planning (FP) workers, and access to FP information in the media.

Duration of abortion law was measured in years – the number of years the current abortion law has been in place. Age was measured also in years of women, and number of living children referred to the total number of children alive at the time of the survey. The rest were measured as follows; marital status (married/cohabiting, never in a union, separated/divorced/widowed); education level (no formal education, primary education, secondary education, higher education); and religion (Christianity, Islamic, Others). Women were considered to access to FP information if they heard of any FP products from the media in the past few months. Those who were also visited by FP worker in the past 12 months were considered to have been visited by FP worker. Health insurance coverage referred to women who had a valid health insurance at the time of the survey. Wealth index was categorized into poorest, poorer, middle, rich, and richest whereas place of residence was classified into urban and rural. The variable selection for this study was based on prior studies in SSA [21,32,33].

## Statistical analyses

Descriptive statistics of the study population were conducted, and were presented in frequencies and percentages for categorical variables and mean for continuous variables. We also presented country-level statistics on year of survey, and sample size of each country (S2 Appendix).

We conducted multilevel logistic regression analyses with four models for each of the outcome variables (modern and LARC/permanent contraceptive use) – null model, model with individual-level covariates only, model with country-level

covariates only, and full model with both individual and country-level covariates. The null model contained only the outcome variable, whereas in the individual-level model only, we controlled for age, place of residence, educational level, wealth index, religion, visit by FP worker, heard of FP in the media, health insurance coverage, and marital status. In the country-level only, we controlled for duration of abortion law years and CHE as a % of GDP. Then in the full model, we controlled for both individual and country-level covariates combined. The same covariates were controlled for in each outcome with the primary exposures being country-level abortion law and adolescent contraceptive legislation.

Intra-cluster correlation coefficient, Akaike Information Criterion, and Bayesian Information Criterion were estimated after fitting each model. The full models performed better than the first three models for each outcome; therefore, we report only the fully-adjusted results. Detailed multilevel modelling results are found in S3 Appendix and S4 Appendix. The multivariable estimates were presented in adjusted odds ratios (aOR) at 95% confidence interval. All covariates were included in the full models irrespective of their significance level at the bivariate models. Sampling weights, clustering, and stratification were incorporated at all levels of the analyses. The analyses were conducted using STATA 17.0 (StataCorp, College Station, Texas, USA).

### Sensitivity analyses

Sensitivity analyses were conducted to assess robustness of the results. First, we estimated the association of restrictive abortion law on each outcome using multilevel linear probability models. Second, we performed multivariable imputation by chained equations (MICE) to account for the missing data because some importance covariates such religion had missingness of greater 10%. Details of MICE analyses is found in S6 Appendix. Third, we assessed whether our results were sensitive when we restricted our study sample to only married women. In addition, we rescaled the data to give equal weights to each of the 31 countries and specified multilevel logistic regression models for each of the two outcome variables. Lastly, we estimated the association of restrictive abortion law on each outcome using logistic regression mixed effects models.

### Ethics approval and informed consent

Ethics approval and informed consent were not required by the institutional review board at Oregon State University. This is because ethical approval for the original Demographic and Health Surveys were provided by the ICF Institutional Review Board, and additional approval was not needed for this secondary analysis of the data, as it used publicly available, de-identified data.

## Results

### Socio-demographic characteristics of study participants

The analyses included a weighted sample of 453,195 women of reproductive age (15–49 years). About half (49%) of the women lived in countries with highly restrictive abortion laws and 27.2% lived in countries with no legislative support for adolescents to access contraception. About six out of ten (58.9%) women lived rural area, one-third (31.2%) had no education, 35.7% were in the lower two wealth index category, and 60% were Christians. More than nine in ten women had no health insurance coverage, and 63.2% were married or living with their partners. About 40% of the women had heard of family planning information in the media but less than one in ten were visited by a family planning worker in the past one year (Table 1).

### Prevalence of modern and LARC/permanent contraceptive use by country

The overall prevalence of modern contraceptive use among women in the 31 countries was 23% (95% CI: 23%, 23%). The top five countries with the highest modern contraceptive prevalence were Namibia, 50% (95% CI: 49%, 51%);

**Table 1. Socio-demographic characteristics of women aged 15-49 years (weighted n = 453,195).**

| Characteristics | Weighted sample distribution | Weighted percent or mean (SD) |
|---|---|---|
| **Overall** | 453,195 | 100 |
| **Abortion law** | | |
| Broadly liberal | 53,808 | 11.9 |
| Moderately restrictive | 177,398 | 39.1 |
| Highly restrictive | 221,990 | 49.0 |
| **Duration of abortion law in years** | 453,195 | 26.8 (19.8) |
| **Legislation that allows adolescents to access contraception** | | |
| No legislative support | 119,059 | 27.2 |
| Partial legislative support | 158,716 | 36.3 |
| Full legislative support | 159,706 | 36.5 |
| **CHE as a % of GDP** | 453,195 | 4.84 (1.9) |
| **Age in years** | 453,195 | 28.6 (9.4) |
| **Number of living children** | 453,195 | 2.4 (2.3) |
| **Place of residence** | | |
| Urban | 186,200 | 41.1 |
| Rural | 266,995 | 58.9 |
| **Educational level** | | |
| No education | 141,195 | 31.2 |
| Primary education | 134,892 | 29.8 |
| Secondary education | 151,122 | 33.3 |
| Higher education | 25,959 | 5.7 |
| **Wealth index** | | |
| Poorest | 78,257 | 17.3 |
| Poorer | 83,463 | 18.4 |
| Middle | 87,714 | 19.4 |
| Richer | 96,174 | 21.2 |
| Richest | 107,588 | 23.7 |
| **Religion** | | |
| Christianity | 242,153 | 60.0 |
| Islamic | 138,808 | 34.4 |
| Others | 22,704 | 5.6 |
| **Visited by FP worker in the past 12 months** | | |
| Yes | 29,581 | 7.5 |
| No | 366,955 | 92.5 |
| **Heard of FP in media in the past few months** | | |
| Yes | 173,645 | 39.8 |
| No | 263,052 | 60.2 |
| **Health insurance coverage** | | |
| Yes | 32,453 | 7.8 |
| No | 384,957 | 92.2 |
| **Marital Status** | | |
| Married/living with partner | 286,436 | 63.2 |
| Never in union | 127,840 | 28.2 |
| Widowed/divorced/separated | 38,919 | 8.6 |

n: Sample size; SD: Standard Deviation; GDP: Gross Domestic Product; CHE: Current Health Expenditure; FP: Family Planning. Missingness; religion = 10.9%, visited by FP worker = 12.5%, health insurance = 7.8%, heard of FP in media = 3.6%.

Lesotho, 49% (95% CI: 47%, 50%); South Africa, 48% (95% CI: 47%, 49%); Zimbabwe, 48% (95% CI: 47%, 49%); and Malawi, 45% (95% CI: 45%, 46%). Seventeen of the countries reported modern contraceptive prevalence below the overall average. There was high heterogeneity in the prevalence of modern use across all countries ($I^2 = 99.9\%$, p value<0.001) (Fig 1A). In terms of the prevalence of LARC/permanent contraceptive use, the overall was 6% (95% CI: 6%, 6%). Malawi, 18% (95% CI: 18%, 19%); Burkina Faso, 15% (95% CI: 14%, 16%); and Kenya, 12% (95% CI: 11%, 12%) reported the top three prevalence. There was also high heterogeneity on LARC/permanent contraceptive prevalence across all countries ($I^2 = 99.7\%$, p value<0.001) with majority of the countries reporting prevalence below the overall average (Fig 1B).

## Association between independent variables and contraceptive use

In the adjusted model, there was 0.62 (95% CI: 058, 0.65) and 0.88 (95% CI: 0.83, 0.93) lower odds of using modern contraceptives among women in countries with moderately and highly restrictive abortion laws compared to those in countries with broadly liberal abortion laws. Furthermore, women in countries with moderately restrictive abortion laws were 0.60 (95% CI: 0.55, 0.66) times as likely to use LARC/permanent contraceptives than those in countries with broadly liberal abortion laws. Women in countries with full legislative support for adolescents to access contraception without parental and spousal consent were 1.55 (95% CI: 1.47, 1.63) and 2.30 (95% CI: 2.14, 2.48) times as likely to use of modern and LARC/permanent contraceptives respectively, than those in countries with no legislative support for adolescent contraceptive use (Table 2).

An increase in CHE as a percent of GDP, increase in age, and increase in the number of living children were all associated with higher odds of modern and LARC/permanent contraceptive use. Women in rural area had lower odds of using modern contraceptives compared to those in urban area but had higher odds of using LARC/permanent contraceptives compared to women in urban area. Furthermore, women with primary education or higher, or poorer wealth index or upper, had higher odds of using modern contraceptives. Also, Islamic women had lower odds of using of modern contraceptives than Christian women (Table 2).

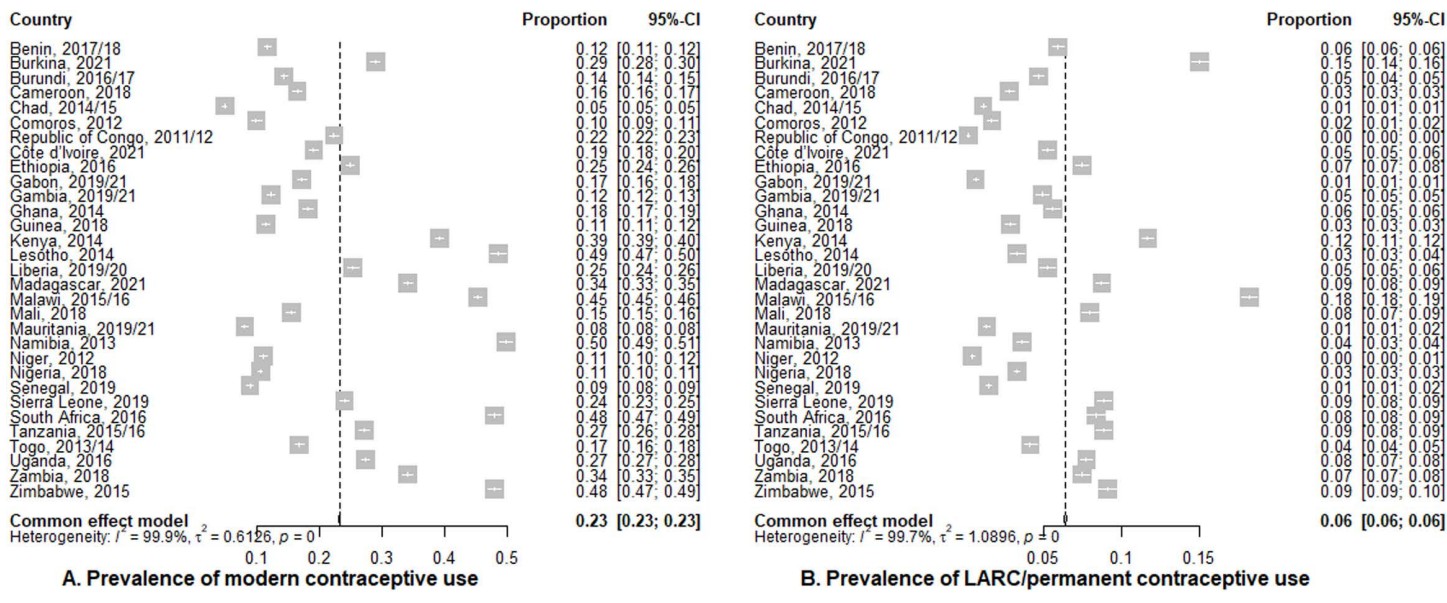

**Fig 1. Prevalence of contraceptive use by county.**

**Table 2. Multivariable logistic regression analysis between independent variables and use of contraception.**

| Characteristics | Currently using modern contraceptive method<br>Adjusted Odds Ratios (95% CI) | Currently using LARC/permanent method<br>Adjusted Odds Ratios (95% CI) |
|---|---|---|
| **Abortion law** | | |
| Broadly liberal | Ref. | Ref. |
| Moderately restrictive | 0.62 (0.58, 0.65)** | 0.60 (0.55, 0.66)** |
| Highly restrictive | 0.88 (0.83, 0.93)** | 1.00 (0.92, 1.09) |
| **Legislation that allows adolescents to access contraception** | | |
| No legislative support | Ref. | Ref. |
| Partial legislative support | 1.04 (0.99, 1.09) | 1.14 (1.06, 1.22)** |
| Full legislative support | 1.55 (1.47, 1.63)** | 2.30 (2.14, 2.48)** |
| **Duration of abortion law in years** | 0.99 (0.99, 0.99)** | 1.00 (1.00, 1.00)* |
| **CHE as a % of GDP** | 1.35 (1.34, 1.37)** | 1.25 (1.23, 1.27)** |
| **Age in years** | 0.98 (0.97, 0.98)** | 0.99 (0.99, 0.99)** |
| **Number of living children** | 1.16 (1.16, 1.17)** | 1.10 (1.18, 1.20)** |
| **Place of residence** | | |
| Urban | Ref. | Ref. |
| Rural | 0.92 (0.88, 0.96)** | 1.15 (1.08, 1.22)** |
| **Educational level** | | |
| No education | Ref. | Ref. |
| Primary education | 2.04 (1.96, 2.12)** | 1.41 (1.34, 1.49)** |
| Secondary education | 2.41 (2.31, 2.52)** | 1.38 (1.29, 1.46)** |
| Higher education | 2.89 (2.70, 3.09)** | 1.54 (1.40, 1.69)** |
| **Wealth index** | | |
| Poorest | Ref. | Ref. |
| Poorer | 1.26 (1.21, 1.32)** | 1.28 (1.20, 1.36)** |
| Middle | 1.40 (1.34, 1.47)** | 1.45 (1.36, 1.55)** |
| Richer | 1.49 (1.40, 1.54)** | 1.58 (1.47, 1.69)** |
| Richest | 1.34 (1.26, 1.41)** | 1.84 (1.70, 1.99)** |
| **Religion** | | |
| Christianity | Ref. | Ref. |
| Islamic | 0.43 (0.42, 0.45)** | 0.52 (0.49, 0.55)** |
| Others | 1.01 (0.94, 1.09) | 0.90 (0.81, 1.00) |
| **Visited by FP worker in the past 12 months** | | |
| Yes | Ref. | Ref. |
| No | 0.76 (0.73, 0.80)** | 0.89 (0.84, 0.94)** |
| **Heard of FP in media in the past few months** | | |
| Yes | Ref. | Ref. |
| No | 0.75 (0.73, 0.77)** | 0.75 (0.72, 0.78)** |
| **Health insurance coverage** | | |
| Yes | Ref. | Ref. |
| No | 1.06 (1.01, 1.12)* | 1.03 (0.95, 1.11) |
| **Marital Status** | | |
| Married/living with partner | Ref. | Ref. |
| Never in union | 0.25 (0.24, 0.26)** | 0.24 (0.22, 0.26)** |
| Widowed/divorced/separated | 0.57 (0.54, 0.59)** | 0.73 (0.68, 0.78)** |

*p value<0.05,

**<0.01. CI: Confidence Interval; FP: Family Planning; GDP: Gross Domestic Product; CHE: Current Health Expenditure.

## Sensitivity results

Results from the sensitivity analyses were similar to the main results. The MICE models, the multilevel linear probability models, and the mixed effects logistic regression models showed that women in countries with moderately and highly restrictive abortion laws had lower use of modern and LARC/permanent contraceptives than those in countries with broadly liberal abortion laws. We found similar results when we rescaled the data to give equal weights to each country and when we also restricted the sample to only married women. Additionally, in all the sensitivity analyses, women in countries with full legislation that supported adolescents access to contraception had higher odds of using modern and LARC/permanent contraceptives than those in countries with no such policies (S5–S9 Appendices).

## Discussion

### Key findings

The study assessed the association of restrictive abortion law with modern contraceptive use among women of 31 SSA countries. One in four women used modern contraceptives whereas fewer than one in ten used LARC/permanent contraceptives. We also found that restrictive abortion laws were associated with lower modern contraceptive use whereas supportive adolescent contraceptive legislations were associated with higher contraceptives use.

Our findings add to the growing evidence that modern contraceptive use among woman of reproductive age in SSA is low [3,27,34,35]. More worrying is the finding that fewer than one in ten women in the study used highly effective contraceptive methods (LARC/permanent methods). Given the high efficacy of LARC/permanent methods in preventing unintended pregnancies, many women may not be benefiting from these methods and thus are at risk of unintended pregnancy. This finding aligns with a recent report by the United Nations that majority of women in SSA who use contraceptives rely on short-acting methods such as condoms, pills, and injectable [27]. Improving modern contraceptive use requires expanding access to LARC/permanent methods, integrating services into primary healthcare, training providers for culturally appropriate counselling and services, implementing community education to dispel myths, and involving male partners, families and community leaders [36–39]. We also observed that contraceptive use varied widely across countries in SSA. For example, although 14 of the countries in this study had an average modern contraceptive use above the SSA average, a substantial number of countries averaged less than 15%. Our finding may reflect the differences in countries commitment to contraceptive promotion as well as individual women preferences to these methods [40].

Women in countries with moderately and highly restrictive abortion laws had lower contraceptive use than those in countries with broadly liberal abortion laws. Prior research suggested that abortion would be used as a substitute for contraceptives, especially when access to contraceptive methods are limited [41–43]. This assumption implicitly assumed that liberalization of abortion laws could lead to lower use of contraceptives, especially in SSA where contraceptive use is lower compared to other regions of the world. It is important to note, however, that contraceptives are not designed to terminate a pregnancy but rather prevent an unintended pregnancy, whereas abortion is used to terminate an existing pregnancy. Thus contraception and abortion complement, rather than substitute, for each other. Furthermore, prior evidence suggests that women would prefer to use contraception and prevent a pregnancy rather than have an abortion [44–46].

Countries with restrictive abortion laws may not be proponents of contraception, and therefore, may not commit to implementing effective contraceptive programs [25,26]. Lack of contraceptive programs may be associated with lower access and subsequently lower use of contraceptives among women in those countries. Furthermore, prior evidence suggested that access to legal abortion care was also associated with postabortion contraceptive use [47–49]. This finding implies that women in countries with restrictive abortion laws may be resorting to unsafe abortion practices without having access to postabortion contraception, likely decreasing their use of contraceptives.

Although studies on abortion restriction and contraceptive use in SSA are limited, findings for studies elsewhere are mixed. In Dominican Republic, a recent difference-in-differences study showed that abortion restriction was associated decrease modern contraceptive use among women of reproductive age [18]. Another cohort study of prescriptions filled at United States pharmacies found that the *Dobbs* decision was associated with declined in oral contraceptives in states enacting highly restrictive abortion laws [15]. Contrary, Medoff found that abortion restriction was association with greater use of highly effective contraceptives, while Felkey & Lybecker found no association between abortion restriction and greater of use of highly effective contraceptives among women in United States [16,17]. Different reasons could explain why abortion restriction may impact contraceptive use in different directions. Some scholars posited that women use more effective contraceptives to prevent unintended pregnancy in highly restrictive abortion settings [17], while other have suggested that abortion restrictions undermine access to comprehensive reproductive health care including contraceptive awareness creation and reduced utilization [16,19].

It is noteworthy that while women in countries with moderately restrictive abortion laws had lower use of LARC/permanent methods compared to those in countries with broadly liberal abortion laws, we observed no such difference between women in countries with highly restrictive and broadly liberal abortion laws. This contrasting finding is unexpected. However, we think that in settings where abortion is completely banned, the inability to access abortion in cases of unintended pregnancy may drive women to use highly effective contraceptives, such as LARC/permanent methods as a means to prevent unwanted pregnancies, even when these methods are in limited supply [17].

We also found that women in countries that had contraceptive policies to support adolescents access to contraception without parental and/or spousal consent had higher use of modern contraceptives than those in countries with no laws to support adolescent contraceptive access. This finding is supported by prior studies that suggested that women's autonomy play an important role in use of contraceptives [50,51]. Reproductive health decisions are often made for many married and unmarried adolescent women in SSA by their male spouses or parents [52]. Most often, spouses and parents discourage or prevent adolescent women from using contraceptives for various reasons [53,54]. However, in countries where they are laws backing use of contraceptives by married and unmarried adolescent women without the approval from a spouse or parent, their contraceptive use will likely increase thereby increasing the overall contraceptive use of women in those countries. In addition, countries that had favorable adolescent contraceptive policies would likely also have favorable contraceptive policies in general, hence the increased use of contraceptives. Our finding underscore the need for policy makers in SSA countries to consider implementing favorable contraceptive policies to promote the use of contraceptives.

Furthermore, the use of modern contraceptives among women was higher in countries with higher CHE as a percentage of GDP. It may be that countries that had higher health expenditure also spent more on reproductive health to improve accessibility and affordability of contraceptive commodities. This possibility may explain why women in those countries had higher use of contraceptives. However, Moreira et al. found a null association between CHE as a percentage of GDP and use of contraception in their study of 14 low-income countries in Latin America and the Caribbean [55]. Whereas their study was restricted to only sexually active women (had sex in the past 30 days prior to the survey), this current study encompassed all women of reproductive age irrespective of their sexual activity status and might, therefore, explain the differences in findings.

## Strengths and limitations

To the best of our knowledge, this is the first study to systematically examined how abortion restrictions impact contraceptive use among women of reproductive age in SSA. The study analyzed nationally representative surveys from 31 countries using analytical variables based on a robust conceptual framework. Our findings are also robust to different model specifications including multivariable imputation by chained equations. These findings add evidence to the long held view that restrictive abortion laws contribute to poor reproductive health service utilization and health outcomes of women in SSA [3,4].

The study had some limitations. The ordinal level of abortion restriction was measured based on the Center for Reproductive Rights [1]. Abortion is permitted on other additional grounds such as rape, incest, and fetal anomaly in some of the countries in this study. These grounds do not affect the overall ordinal categorization by Center for Reproductive Rights although they can be significant in expanding legal abortion access in countries with highly or moderately restrictive laws [1]. Despite this limitation, the CRR classification is an important measure for abortion restriction and has been used in the literature including analyses from the Guttmacher Institute [3]. Another limitation of this study was missing data. Data on some key variables such as religion were not available for South Africa, Tanzania, Mauritania and Niger. This limitation was addressed by conducting multivariable imputation analysis, and the results were similar to the main findings. Furthermore, we did not control for country's contraceptive supply in our models because such data were unavailable. Supply of contraceptives is one of the important predictors of contraceptive use [40,56], and could have potentially caused omitted variable bias. However, we expect this omission to be minimal because we controlled for country's adolescent contraceptive policy and individual level factors such as visits by family planning worker and access to contraceptive information in the media. These variables are good proxies for country's contraceptive supply level. Lastly, causality cannot be inferred from our study because we used cross sectional data for the analysis.

## Policy implications

Our study showed that 88% of women of reproductive age lived in countries with moderately and highly restrictive abortion laws. Given that restrictive abortion laws are associated with lower use of contraceptive methods, women in countries with restrictive abortion laws face a double challenge of limited access to contraceptives as well as legal abortion services. Our study contributes to the overwhelming evidence that abortion restriction is not only associated unsafe abortion and its complications [3,4,6], but it may also be associated with lower use of modern contraceptives. The findings imply that abortion restriction may be contributing to the high incidence of unintended pregnancies in SSA. Prior research suggested that unintended pregnancy may induce psychological distress among women with detrimental effects on birth outcomes [7,57–59]. Therefore, it is evident from our findings as well as prior research that liberalization of abortion laws and implementation of effective contraceptive programs in SSA could play significant roles towards improving reproductive health indicators in the region.

It is also crucial to acknowledge that socio-economic factors such age, marital status, education, household wealth, and place of residence play a significant role in shaping reproductive health service utilization and health outcomes of women [21,60–62]. It is, therefore, important to consider and address these factors when reforming abortion laws. Socio-economic disparities can be reduced when policy makers deliberately design interventions that factor in the needs of the disadvantaged in society.

## Conclusion

Findings from this study show that restrictive abortion law was associated with lower use of modern contraceptives among women in SSA. In contrast, contraceptive policies that supported adolescents access to contraceptives without parental and/or spousal consent were associated with higher use of contraceptives among women. Based on these findings, abortion law liberalization together with expanded reproductive health services (e.g., supply of contraceptive methods and services) in SSA are recommended. These changes have the potential to improve access to and use of contraceptives among women in SSA.

## Supporting information

**S1 Appendix. Flow chart for inclusion and exclusion criteria for countries in the study.**
(DOCX)

**S2 Appendix. Country-level descriptive statistics, weighted n = 453,195.**
(DOCX)

**S3 Appendix. Multilevel modeling for modern contraceptive use.**
(DOCX)

**S4 Appendix. Multilevel modeling for LARC/permanent contraceptive use.**
(DOCX)

**S5 Appendix. Sensitivity analysis.**
(DOCX)

**S6 Appendix. Sensitivity analysis.**
(DOCX)

**S7 Appendix. Sensitivity analysis.**
(DOCX)

**S8 Appendix. Sensitivity analysis.**
(DOCX)

**S9 Appendix. Sensitivity analysis.**
(DOCX)

**S1 Data: Data for manuscript.**
(XLSX)

## Acknowledgments

We thank the Demographic and Health Surveys for giving us access to the data for this study.

## Author contributions

**Conceptualization:** Maxwell Tii Kumbeni, Marit L. Bovbjerg, S. Marie Harvey, Chunhuei Chi, Jeff Luck.

**Data curation:** Maxwell Tii Kumbeni, Jeff Luck.

**Formal analysis:** Maxwell Tii Kumbeni.

**Methodology:** Maxwell Tii Kumbeni, Marit L. Bovbjerg, Jeff Luck.

**Supervision:** Marit L. Bovbjerg, S. Marie Harvey, Chunhuei Chi, Jeff Luck.

**Writing – original draft:** Maxwell Tii Kumbeni, Marit L. Bovbjerg, S. Marie Harvey, Chunhuei Chi, Jeff Luck.

**Writing – review & editing:** Maxwell Tii Kumbeni, Marit L. Bovbjerg, S. Marie Harvey, Chunhuei Chi, Jeff Luck.

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
