## [Decision Letter · Decision Letter 0]

PGPH-D-25-00843

The role of restrictive abortion laws on modern contraceptive use in Sub Saharan Africa

Dear Dr. Kumbeni,

Thank you for submitting your manuscript to PLOS Global Public Health. After careful consideration, we feel that it has merit but does not fully meet PLOS Global Public Health’s publication criteria as it currently stands. Therefore, we invite you to submit a revised version of the manuscript that addresses the points raised during the review process.

The reviewers have recommended a minor revision. Please ensure that you provide a point-to-point response to reviewer's comments as a separate document. Highlight the changed in your main manuscript either as tracked chaneges or highlighting the speicfic changes in the manuscript by a different color. 

We look forward to receiving your revised manuscript.

Kind regards,

Tanmay Bagade, Ph.D., MS (O&G), MPH, MHM

Academic Editor

Journal Requirements:

1. We noticed you have some minor occurrence of overlapping text with the following previous publication(s), which needs to be addressed:

-https://www.guttmacher.org/report/from-unsafe-to-safe-abortion-in-subsaharan-africa

In your revision ensure you cite all your sources (including your own works), and quote or rephrase any duplicated text outside the methods section. Further consideration is dependent on these concerns being addressed.

Additional Editor Comments (if provided):

Reviewers' comments:

Reviewer's Responses to Questions

**Comments to the Author**

1. Does this manuscript meet PLOS Global Public Health’s publication criteria ? Is the manuscript technically sound, and do the data support the conclusions? The manuscript must describe methodologically and ethically rigorous research with conclusions that are appropriately drawn based on the data presented.

Reviewer #1: Yes

Reviewer #2: Yes

Reviewer #3: Yes

2. Has the statistical analysis been performed appropriately and rigorously?

Reviewer #1: Yes

Reviewer #2: Yes

Reviewer #3: Yes

3. Have the authors made all data underlying the findings in their manuscript fully available (please refer to the Data Availability Statement at the start of the manuscript PDF file)?

Reviewer #1: Yes

Reviewer #2: Yes

Reviewer #3: Yes

4. Is the manuscript presented in an intelligible fashion and written in standard English?

Reviewer #1: Yes

Reviewer #2: Yes

Reviewer #3: Yes

5. Review Comments to the Author

Reviewer #1: The manuscript is technically sound and is written in clear and understandable language following the standard research format. Methodology details like data source, study design, setting, participants, variable were followed. The study used robust and rigorous statical analysis and the sensitivity analyses indicates a strong focus on assessing the robustness and reliability of the study findings. The results are presented clearly and the discussion are drawn based on the data presented. The authors also acknowledge limitations.

NOTE: Please crosscheck and verify L 250-251 (there ware two separate figures for Zimbabwe)

Reviewer #2: 1. A more specific recommendations specially in the abstract section regarding what type of abortion law reforms and favorable contraceptive policies would be needed will increase interest and application of the recommendations.

2. Under the discussion section apart from the reforms and policies, it would be great to add other measures should be taken to increase acceptance and use of the highly effective contraceptive methods (LARC/permanent

methods). E.g. community engagement and working with key influencers such as men, religious leaders etc.

Reviewer #3: This manuscript examines the impact of restrictive abortion laws on modern contraceptive use. The authors utilize secondary data from Demographic and Health Surveys (DHS) across 31 Sub-Saharan African (SSA) countries, combined with the 2022 abortion law classification data from the Center for Reproductive Rights (CRR). They apply rigorous analytical methods, appropriately addressing missing data, adjusting for potential confounders, and conducting sensitivity analyses to assess the robustness of the results. The language is generally clear and appropriate for scientific communication.

There are few areas for revision.

Abstract:

1) Line # 36: 058 should be 0.58 in the confidence interval

Results

1) Page 11, Table 1, Higher education row: Please check the number and the percentage. There must be something wrong. Any missing digits?

2) Page 13, Table 2, Abortion law row: For highly restrictive abortion laws and current use of LARC/permanent methods, the confidence interval crosses the null value, suggesting no statistically significant association. This contrasts with the significant association observed between moderately restrictive laws and modern contraceptive use. This discrepancy should be explicitly addressed in the discussion section.

3) Line # 302 to 304 (page 15): please check the language and edit it. There must be a typo here.

6. PLOS authors have the option to publish the peer review history of their article (what does this mean? ). If published, this will include your full peer review and any attached files.

**Do you want your identity to be public for this peer review?** For information about this choice, including consent withdrawal, please see our Privacy Policy .

Reviewer #1: No

Reviewer #2: **Yes: ** Dr Asrat Dibaba Tolossa

Reviewer #3: No

---

## [Editor Report · Decision Letter 1]

The role of restrictive abortion laws on modern contraceptive use in Sub Saharan Africa

PGPH-D-25-00843R1

Dear Mr. Kumbeni,

We are pleased to inform you that your manuscript 'The role of restrictive abortion laws on modern contraceptive use in Sub Saharan Africa' has been provisionally accepted for publication in PLOS Global Public Health.

Best regards,

Tanmay Bagade, Ph.D., MS (O&G), MPH, MHM

Academic Editor